# Systematic Identification and Comparison of the Expressed Profiles of lncRNAs, miRNAs, circRNAs, and mRNAs with Associated Co-Expression Networks in Pigs with Low and High Intramuscular Fat

**DOI:** 10.3390/ani11113212

**Published:** 2021-11-10

**Authors:** Feng Cheng, Jing Liang, Liyu Yang, Ganqiu Lan, Lixian Wang, Ligang Wang

**Affiliations:** 1Key Laboratory of Farm Animal Genetic Resources and Germplasm Innovation of Ministry of Agriculture of China, Institute of Animal Science, Chinese Academy of Agricultural Sciences, Beijing 100193, China; cf52597@163.com (F.C.); m18738133860@163.com (L.Y.); 2College of Animal Science and Technology, Guangxi University, Nanning 530004, China; liangjing@gxu.edu.cn (J.L.); gqlan@gxu.edu.cn (G.L.)

**Keywords:** pigs, IMF content, whole transcriptome, co-expression networks

## Abstract

**Simple Summary:**

Intramuscular fat (IMF) is a key factor affecting many meat quality traits of pigs, such as pork tenderness, flavor, and many more. In this study, a systematic identification and comparison of the expressed profiles of messenger RNA (mRNAs), long non-coding RNAs (lncRNAs), microRNAs (miRNAs), and circular RNAs (circRNAs) with associated co-expression networks longissimus dorsi muscle (LDM) in Large White × Min pigs F2 resource population were performed. The results contain high-throughput genomic data, which are helpful to clarify the regulatory role of a variety of RNAs in regulating intramuscular fat formation and lipid metabolism at the genomic level and provide new insights for studying the mechanism of fat formation and the regulation of meat quality related genes at the molecular level.

**Abstract:**

Intramuscular fat (IMF) content is a complex trait that affects meat quality and determines pork quality. In order to explore the potential mechanisms that affect the intramuscular fat content of pigs, a Large white × Min pigs F2 resource populations were constructed, then whole-transcriptome profile analysis was carried out for five low-IMF and five high-IMF F2 individuals. In total, 218 messenger RNA (mRNAs), 213 long non-coding RNAs (lncRNAs), 18 microRNAs (miRNAs), and 59 circular RNAs (circRNAs) were found to be differentially expressed in the longissimus dorsi muscle. Gene ontology analysis and Kyoto Encyclopedia of Genes and Genomes annotations revealed that these differentially expressed (DE) genes or potential target genes (PTGs) of DE regulatory RNAs (lncRNAs, miRNAs, and circRNAs) are mainly involved in cell differentiation, fatty acid synthesis, system development, muscle fiber development, and regulating lipid metabolism. In total, 274 PTGs were found to be differentially expressed between low- and high-IMF pigs, which indicated that some DE regulatory RNAs may contribute to the deposition/metabolism of IMF by regulating their PTGs. In addition, we analyzed the quantitative trait loci (QTLs) of DE RNAs co-located in high- and low-IMF groups. A total of 97 DE regulatory RNAs could be found located in the QTLs related to IMF. Co-expression networks among different types of RNA and competing endogenous RNA (ceRNA) regulatory networks were also constructed, and some genes involved in type I diabetes mellitus were found to play an important role in the complex molecular process of intramuscular fat deposition. This study identified and analyzed some differential RNAs, regulatory RNAs, and PTGs related to IMF, and provided new insights into the study of IMF formation at the level of the genome-wide landscape.

## 1. Introduction

Pork is one of the main sources of human protein and fat, accounting for more than 40% of global human meat consumption [1]. Meat quality is an important economic trait in pig production, and it can be evaluated by multiple indicators, such as intramuscular fat content (IMF), muscle tenderness, meat color, and water-holding capacity [2,3]. IMF is the key meat quality trait affecting the tenderness, flavor, and juiciness of pork. Appropriate intramuscular fat content can improve meat quality [4]. Due to the different genetic backgrounds and breeding objectives, there are considerable differences in intramuscular fat content between native Chinese pigs and Western pigs. One famous local pig breed in China, the Min pig, has excellent meat quality, delicious flavor and a high intramuscular fat content (>4%), which provides good research material for the study of gene regulation related to intramuscular fat deposition in pigs.

In the past, many studies have used low-density microsatellite markers to identify QTL associated with porcine IMF [5,6,7]. However, due to the low marker density, it is difficult to accurately locate the target genes [8]. With the emergence of high-throughput genotyping techniques, such as single nucleotide polymorphism (SNP) arrays, IMF-related genetic variations and QTL of pigs can be found in a narrower gene region [9]. Combined with genome-wide association study (GWAS), potential genetic molecules relating to intramuscular fat content can be identified. To date, 786 QTLs have been identified for IMF (https://www.animalgenome.org/cgi-bin/QTLdb/SS/traitsrch?tword=Intramuscular%20fat, release 44, accessed on 26 April 2021). At the same time, an increasing number of messenger RNA (mRNAs) and regulatory RNAs such as long non-coding RNAs (lncRNAs), microRNAs (miRNAs) and circular RNAs (circRNAs) have been identified through sequencing as candidate genes or important regulators of fat deposition or lipid metabolism in pigs [9,10,11]. However, combined analyses of all these types of RNA have rarely been reported [12], and an in-depth functional analysis of regulatory RNAs for IMF development in pigs has not yet been conducted.

In this study, we used whole-transcriptome sequencing to investigate the differences in the expression of mRNAs, lncRNAs, miRNAs, and circRNAs between low- and high-IMF in longissimus dorsi muscle (LDM) in a Large White × Min pigs F2 resource population. Functional analysis of mRNAs, regulatory RNAs, and the potential target genes (PTGs) of regulatory RNAs was performed to analyze the function of differential expressed RNAs (DERs). The differential expressed (DE) regulatory RNAs were then mapped onto the QTL database to predict their function. Finally, the co-expression networks of regulatory RNAs were also explored to filter the candidate RNAs related to IMF.

## 2. Materials and Methods

### 2.1. Ethics Statement

All animals used in this study were handled and kept according to the standard guidelines for experimental animals established by Ministry of Science and Technology (Beijing, China). All animal experiments were carried out with the ethical approval (No. IAS2020-109) of the Animal Ethics Committee of the Institute of Animal Science, Chinese Academy of Agricultural Sciences.

### 2.2. Animal and Sample Preparation

In this study, 10 individuals were selected from the F2 population of Large White × Min pigs (at slaughter the average age was 240 ± 7 days). These pigs were raised in the same environment with the same feeding conditions. Pigs were weighed and slaughtered in a commercial slaughterhouse. Tissue samples were collected from the same position (10th to 11th ribs) of the longissimus dorsi muscle of pigs, and then frozen in liquid nitrogen and stored in a refrigerator at −80 °C for further analysis. The IMF content was measured using Soxhlet extractor method following the standard guidelines of the US National Pork Producers Council (NPPC). Ten individuals (five individuals in each group) from two groups were selected according to their content for transcriptome analysis. Table 1 shows the carcass weight, IMF content and grouping of the research samples. The average IMF content of F2 population was 2.85 ± 1.83 and the average carcass weight was 109.31 ± 16.07 kg. The individuals in the study were from the F2 population, the high-IMF group: 4.07 < IMF < 5.43, low-IMF group: 1.05 < IMF < 1.60, and had an average carcass weight of 101.72 ± 12.89 kg. There was significant differences of IMF content between the two groups in this study (*p* < 0.01).

### 2.3. Construction and Sequencing of cDNA Libraries

Total RNA from the longissimus dorsi muscle tissue of each individual was extracted using TRIzol reagent (Invitrogen, Waltham, MA, USA). Bioanalyzer 2100 (Agilent Technologies, Inc., Santa Clara, CA, USA), Nano 6000 Assay Kit (Agilent Technologies, Inc., Santa Clara, CA, USA) and 1% agarose gel electrophoresis were used to determine the quality and integrity of the RNA. The OD 260/280 ratio of the samples was between 1.9 and 2.0, and the RNA integrity of all samples exceeded 7.4. The Ribo-Zero Gold kit (Epicentre, Madison, WI, USA) was used to remove ribosomal RNA from each sample. Two libraries were designed for whole-transcriptome sequencing, miRNAs analysis was used to construct a small RNA library, and lncRNA/circRNA analysis was used to construct a ribosome-removed library. The Agilent DNA 1000 kit on a Bioanalyzer 2100 (Agilent Technologies, Inc.) was used to examine the size and purity of each cDNA library. Finally, these libraries were sequenced on the Illumina HiSeq 4000 (Illumina, San Diego, CA, USA) platform to obtain paired-end reads.

### 2.4. Data Mapping and Transcriptome Assembly

Content that contained poly-N or adapters and low-quality reads was deleted from the sequenced row data, and the remaining reads were called clean data. The clean data were mapped to the pig reference genome (*Sus scrofa* 11.1) using the default parameters of HISAT (v2.0.4) software [13], and the mapped reads of each sample had at least one of two replicates. The transcripts were assembled and annotated using the default parameters in StringTie (v1.3.1) software [14]. Using Bowtie (v1.0.0) software [15], sequence alignment was performed for the Silva database (http://www.arb-silva.de/, version: 138.1, accessed on 2 June 2021), the GtRNAdb database (http://lowelab.ucsc.edu/GtRNAdb/, version: SGSC Sscrofa9.2, accessed on 5 June 2021), the Rfam database (http://rfam.xfam.org/, version: Rfam 14.6, accessed on 6 June 2021) and the Repbase database (http://www.girinst.org/repbase/, version: RepBase26.10, accessed on 6 June 2021). Non-codingRNAs (ncRNAs) and repeat sequences such as ribosomal RNA (rRNA), transfer RNA (tRNA), small nuclear RNA (snRNA), and small nucleolar RNA (snoRNA) were screened to construct the small RNA (sRNA) sequence information.

### 2.5. Identification of lncRNAs, miRNAs, and circRNA

The basic screening conditions of transcript information were as follows: (1) we selected the transcripts whose class code was ‘i’, ‘x’, ‘u’, ‘o’, or ‘e’; (2) we selected the transcripts whose length was ≥200 bp with an exon number ≥ 2; and (3) we selected the transcripts with a Fragments per kilobase of exon model per million mapped fragments (FPKM) ≥ 0.1. CPC2 (CPC2-beta) [16], CNCI (v2) [17], and CPAT [18] software were used to predict the coding potential from the basically screened transcripts. The lncRNA retained in the four databases was defined as new lncRNAs, and the predicted lncRNAs were classified. The reads of the reference genome were compared with the mature sequences of known miRNAs in the miRBase database (https://www.mirbase.org/search.shtml, version: Release 22.1, accessed on 22 July 2021) and the range from 2 nt upstream to 5 nt downstream to identify known miRNAs. In addition, the miRDeep2 (v2.0.5) software package [19] was used to predict new miRNAs based on the distribution information of reads on the precursor sequences and the energy information of the precursor structure. The Sam alignment was scanned twice by CIRI (v2.05) [20] software based on the BWA-MEM algorithm to detect junction reads with paired chiastic clipping signals. These comparisons were then scanned again using dynamic programming algorithms to filter the false positive candidates caused by error mapping reads. Finally, circRNA was identified by reading at least two connections. 

### 2.6. Differentially Expressed RNA Analysis

The expression levels of the transcripts were calculated using StringTie and Ballgown software, and standardized using FPKM (fragments per kilobase of transcript per million fragments mapping) and RSEM (splicing reads per billion mapping). StringTie uses FPKM as an indicator to measure the expression level of transcripts or genes (mRNA and lncRNA). SRPBM (splicing reads per billion mapping) was used to estimate the expression level of circRNA. MiRNA expression quantification was normalized by the TPM algorithm. DEseq2 (v1.6.3) R package [21] was used to screen DERs with a fold change ≥ 1.5 and a *p*-value < 0.01 (mRNA, lncRNA, and circRNA), or a fold change ≥ 1.5 and *p*-value < 0.05 (miRNA).

### 2.7. Prediction of the Potential Target Genes of DE lncRNAs, miRNAs, and circRNAs

In this study, two strategies were used to predict lncRNA target genes: (1) cis-target gene prediction based on the position of the lncRNA and the target gene located upstream or downstream (<100 Kb) from the lncRNAs; (2) trans-target gene prediction based on the correlation analysis of lncRNA and mRNA expression, with the genes identified as PTGs of lncRNAs when these distant protein-coding genes were positively or negatively correlated with the expression of lncRNAs. The absolute Pearson’s coefficient (r) between each lncRNA and protein-coding gene pair was ≥0.95, and the *p*-value was <0.01 [22]. MiRNA target genes were predicted using miRanda (v3.3a) [23] and Targetscan [24] software; the intersection of the two target prediction results was taken as the miRNA target gene. The gene corresponding to the longest transcriptional fragment that accurately matched both ends (5′ end or 3′ end) of the circRNA was used as the host gene of the circRNA.

### 2.8. Gene Ontology Enrichment and KEGG Pathway Analyses

Gene ontology (GO) enrichment analysis and Kyoto Encyclopedia of Genes and Genomes (KEGG) pathway enrichment analysis were performed for all DERs between the two groups. GO (http://www.geneontology.org/, version: Release 2021-10-26, accessed on 28 July 2021) is the international standard classification of gene functions. It classifies gene functions according to three aspects: molecular function, biological processes and cell composition. The KEGG (http://www.genome.jp/kegg, version: Release 99.1, accessed on 28 July 2021) database is the main public database for metabolic analysis and regulation network research. In order to explore the main biological functions of differentially expressed genes on the basis of hypergeometric distribution, clusterProfiler (v3.10.1) [25] was used for GO and KEGG signal pathway enrichment analysis of mRNA. GO terms and pathways with *p* < 0.05 were considered to be significantly enriched.

### 2.9. Co-Construction of Gene Expression Networks

According to the RNA expression data, Pearson’s correlation analysis was used to construct a co-expression network of mRNA–lncRNA, mRNA–circRNA, circRNA–lncRNA, and circRNA–miRNA pairs with thresholds |r| > 0.8 and *p* < 0.05. In addition, the competing endogenous RNA (ceRNA) regulatory network was constructed on the basis of the pairwise expression results of the different RNAs. At the same time, a one-step neighbor network of differential RNA was extracted from each differential combination in the ceRNA relationship pair, and the differential ceRNA relationship pair was obtained. Based on a random walk, the key nodes in the ceRNA network were sorted, and the top 5% RNAs in the network were screened as key genes. Functional annotation, pathway enrichment analysis, and network construction of the key genes were carried out. 

### 2.10. Association Analysis between QTL Sites and the Locations of Differentially Expressed RNA 

For the combined analysis of DERs and QTLs, the data containing the location of the DERs were compared with the filtered pig QTL data. Bedtools (v2.27.1) [26] software was used and the ‘intersection’ command was used: intersectBed-a-b-wa-wb.

### 2.11. Validation of the RNA Sequencing Results Using qRT-PCR

Three RNA samples each from the two groups were used for qRT-PCR to verify the data of the RNA-seq sequencing results. The cDNA chain was synthesized using the PrimeScript RT reagent Kit with the gDNA Eraser (Takara, Otsu, Japan), and the concentration and quality were determined using a Nanodrop 2000 spectrophotometer (Thermo Fisher Scientific, Waltham, MA, USA). Next, TB Green Premix Ex Taq (Takara) was used for qRT-PCR, which was performed on an Applied Biosystems 7300 Real-Time PCR System (Thermo Fisher Scientific). The thermal cycle parameters used were as follows: Stage 1: 95 °C for 30 s; Stage 2: 95 °C for 5 s and 60 °C for 34 s for 40 cycles; and Stage 3: 95 °C for 15 s, 60 °C for 1 min and 95 °C for 15 s. The glyceraldehyde-3-phosphate dehydrogenase gene (*GAPDH*) was used as an endogenous control gene. The average ΔCt of the low-IMF group individuals was used as sample controls. All qRT-PCR verifications were performed using three biological replicates and with three replicates for each sample. The relative abundance of transcripts was calculated by the 2^−Δ^^ΔCt^ method. The primers (Appendix A) used for qRT-PCR were designed using Oligo7 software and synthesized by Invitrogen Inc. (Shanghai, China). 

### 2.12. Statistical Analyses

The software packages SPSS (v22.0) [27] and GraphPad Prism (v8.0) [28] were used for data analysis and mapping. The results were expressed as means ± standard deviation (SD). One-way ANOVA was used to determine the statistical differences between any two groups, followed by Tukey’s test for multiple comparisons. *p* < 0.05 was considered to indicate a significant difference; *p* < 0.01 and *p* < 0.001 indicated extremely significant differences.

## 3. Results

### 3.1. Overview of RNA Sequencing

After quality control, 228.75 Gb of clean data were obtained from 10 samples; for each sample, the clean data reached 18.24 Gb, and the Q30 base percentage was above 94.07%. The clean reads of each sample were aligned with the pig reference genome (*Sus scrofa* Sscrofa11.1_102). The total percentage of mapped reads of mRNA and lncRNA in the genome was between 96.28% and 97.21%, and the specific comparison results were between 87.45% and 92.61%. In addition, the matching rate of miRNA was 68.51–76.29%, and the matching rate of circRNA was more than 99%. This is basically consistent with the data in other porcine muscle transcriptome studies. It indicates that the data sequencing quality and comparison rate were high, the data utilization rate was normal, and the data met the needs of subsequent analyses. Details are shown in Appendix A.

### 3.2. Differential Expression Profiles ofmRNAs lncRNAs, miRNAs, and circRNAs

A global display of the differentially expressed RNAs on the chromosomes and the quantitative statistics of the DERs are shown in Figure 1. Top DE genes and ncRNAs are shown in Table 2. The top DE genes, such as secreted phosphoprotein 1 (*SPP1*), myosin heavy chain 7B (*MYH7B*), calcium and integrin binding family member 2 (*CIB2*), and other DE genes, such as secreted frizzled-related protein 4 (*SFRP4*), Glycerol-3-phosphate ethyltransferase (*GPAT*), Acetyl-CoA Acyltransferase 2 (*ACAA2*), Acyl-CoA oxidase 2 (*ACOX2*), thrombospondin 4 (*THBS4*), C-C Motif Chemokine Ligand 4 (*CCL4*), C-C motif chemokine ligand 10 (*CCL10*), C-X-C motif chemokine ligand 10 (*CXCL10*), C-X-C motif chemokine ligand 10 (*CXCL16*), and transforming growth factor beta 3 (*TGFB3*), have known functions associated with muscle or fat traits.

Using a fold change ≥ 1.5 and *p* < 0.01 as the standard for screening DE lncRNA circRNA, and mRNA, 218 differentially expressed genes were found between the high- and low-IMF groups, of which 100 were upregulated and 118 were downregulated. Among the 213 differentially expressed lncRNAs, 148 were upregulated and 65 were downregulated, and of the 59 circRNAs, 36 were upregulated and 23 were downregulated. In addition, according to the criteria of fold change ≥ 1.5 and *p* < 0.05, 18 DE miRNAs were identified between the two groups (Appendix A).

### 3.3. Prediction of the Potential Target Genes (PTGs) of DE lncRNAs, circRNAs, and miRNAs

In order to reveal the potential function of the screened DE lncRNAs in the IMF, independent cis- and trans-algorithms were used to predict the target genes. We predicted the cis-regulated PTGs and obtained 692 PTGs that corresponded to 213 DE lncRNAs; 17 of the 692 PTGs were differentially expressed between the two groups (Figure 2A). We then predicted 6663 PTGs of 209 DE lncRNAs via the trans mode: 166 of the 6663 PTGs corresponded to 40 lncRNAs which were differentially expressed between the two groups (Figure 2B). In addition, 21 of the 40 DE lncRNAs upregulated most of their DE PTGs, and 19 DE lncRNAs downregulated the majority of their DE PTGs. For target gene prediction of the DE miRNAs, 15 of 18 DE miRNAs obtained 8775 PTGs. The number of target genes of known miRNAs was very different from the target genes of novel miRNAs. The known miRNAs ssc-mir-190b and ssc-mir-194a-5p had two and four PTGs, respectively, but the novel miRNA_100 had 1585 PTGs. Moreover, 90 of 8775 PTGs were differentially expressed between the two groups. CircRNA has a unique closed-loop mode, and each circRNA had its corresponding PTG. In total, 59 PTGs were obtained, of which only 52 were annotated. Only one of the 59 PTGs was DE between the two groups. Details are shown in Appendix A.

### 3.4. GO and KEGG Analysis of the DERs

GO analysis showed that the DE mRNAs, lncRNAs, miRNAs, and circRNAs were mainly involved in the cell part of the cellular component category. In the biological process category, cell process, single biological process, and biological regulation were the most abundant. DE mRNA and miRNAs were significantly enriched in cell differentiation system development and animal organ development, involving muscle cell differentiation, system development, and tissue and organ development (Figure 3A,B). The target genes of DE circRNAs were mainly in plasma membrane repair, cerebellar Purkinje cell differentiation, N-glycan processing, skeletal muscle contraction, muscle system processes, etc., in muscle development and cell differentiation, and biological signal responses (Figure 3C).

For DE mRNA, the KEGG pathway analysis showed that these DE mRNAs were mainly related to lipid metabolism, such as the cytokine–cytokine interaction receptor, focal adhesion, and the Toll-like receptor signaling pathways. Genes in the cytokine–cytokine receptor interaction, chemotherapeutic factors (such as *CCL4*), transforming growth factor (such as *TGFB3*), and chemokines (such as *CXCL10* and *CXCL16*) pathways were highly expressed in low-IMF individuals. *CCL4, CCL10*, and *SPP1* were enriched in the Toll-like receptor signaling pathway and were all expressed in low-IMF pigs. In addition, primary bile acid biosynthesis included the fatty acid oxidation gene *ACOX2*, which is involved in the lipid synthesis pathway. These genes had the opposite expression trend in high-IMF individuals (Figure 3A).

For DE miRNAs, the KEGG pathway analysis showed that these DE miRNAs were significantly enriched in aminoacyl-tRNA biosynthesis and axon guidance, and the analysis also found pathways closely related to lipid metabolism, including the glucagon signaling pathway and the mTOR signaling pathway, which were also significantly enriched in lipid metabolism, such as glycerol metabolism. The MAPK signaling pathway, the PI3K-Akt signaling pathway, and the insulin signaling pathway are closely related (Figure 3B). In these two pathways, there were seven new miRNAs, of which novel_miR_398, novel_miR_118, and novel_miR_278 were upregulated, and novel_miR_100, novel_miR_476, and novel_miR_7 were downregulated in both pathways. Novel_miR_434 and novel_miR_45, play unique roles in these pathways (Appendix A). 

For the circRNAs, the KEGG pathway analysis showed that these DE circRNAs were highly enriched in inflammatory bowel disease (IBD), hypertrophic task (HCM), etc. In addition, in the tight junction, the adherens junction was also significantly enriched. Functional annotation found that these pathways were associated with lipid metabolism (such as the MAPK signaling pathway, the TGF-beta signaling pathway and cytokine–cytokine receptor interaction). These pathways play a key role in lipid metabolism and synthesis (Figure 3C). 

### 3.5. Functional Analysis of the DE PTGs

The dot-plot analysis showed the results of the top 20 GO analysis results with *p*-values from smallest to largest, as well as the results of KEGG pathway analysis (Figure 4A,B). Most of the PTGs were related to cell density biological processes, but were also significantly enriched in lipid metabolism processes including lipid transport (Figure 4A). Most KEGG pathways which the PTGs were involved in were autoimmune diseases and hormone signal regulation, among which the cGMP-PKG signaling pathway and the estrogen signaling pathway were closely related to IMF content (Figure 4B). 

GO results based on cis-regulation showed that the fatty acid metabolism process, the regulation of lipid catabolic process, the muscle cell apoptosis process, myotube differentiation involved in skeletal muscle regeneration, and regulation of skeletal muscle fiber development were significantly enriched, and were mainly involved in fatty acid metabolism, the lipid catabolic process, myotube differentiation, and muscle fiber development regulation. In our study, several lncRNA target genes were involved in lipid metabolism: MSTRG.1611.1, MSTRG.35593.1, MSTRG.77761.1, MSTRG.22650.3, MSTRG.2132.1, and MSTRG.20935.1 were highlighted. MSTRG.1611.1 and its target gene, acetyl-CoA acyltransferase 2 (*ACAA2*), were downregulated between the two groups, indicating that MSTRG.1611.1 may regulate fatty acid metabolism by negatively affecting *ACAA2*. The top five significantly enriched biological processes were the fatty acid metabolic process, the allantoin metabolic process, the isoleucine metabolic process, the valine metabolic process and the creatine metabolic process. These terms are linked to genes involved in the network (Figure 4C and Appendix A).

KEGG analysis showed that the cis-regulated PTGs of lncRNAs were significantly annotated in glycerophospholipid metabolism, the phospholipase D signaling pathway and the cGMP-PKG signaling pathway; the latter was involved in lipid and carbohydrate metabolism-related pathways, such as the MAPK signaling pathway and fatty acid biosynthesis. The trans-regulated target genes of lncRNAs were enriched in 59 pathways, some of which are associated with lipid metabolism, such as the cGMP-PKG signaling pathway, the regulation of lipolysis in adipocytes and the PPAR signaling pathway, but these were not the most significantly enriched (Appendix A).

### 3.6. Overlapping Analysis between QTL Sites and the Location of DE RNAs 

In order to explore the function of DE RNAs more accurately, we combined DE RNAs with QTLs. The results showed that 208 DE lncRNAs were related to 13,302 QTLs, and 3275 QTLs related to fat deposition were found. These QTLs were distributed on porcine autosomal and X chromosomes, among which chromosomes 1, 7, 2, and 6 were the most distributed, mainly related to backfat, such as average backfat thickness, final rib backfat and subcutaneous shoulder fat thickness. Moreover, 10.63% (348/3275) of the QTLs were associated with intramuscular fat and distributed on chromosomes 2,3, 4, 6, 7, 8, 9, 13, 15, 17, and X, of which chromosome 3 had the most QTLs (223) and was associated with 72 lncRNAs (Figure 5A). 

There were 17 DE miRNAs distributed in 1083 QTLs, of which 252 QTLs were related to fat deposition and were mainly distributed on chromosomes 1, 2, 4, and 7. Most of these QTLs were related to average backfat thickness, final rib backfat, and subcutaneous shoulder fat thickness. At the same time, the newly predicted miRNA novel_miR_45 and the mature miRNA ssc-miR-190b on chromosome 4 and chromosome 17 were closely related to the QTLs for intramuscular fat content (Figure 5B). For DE circRNAs, 58 circRNAs matched a total of 3970 QTLs, 1028 of which were QTLs related to fat deposition, which were distributed on the autosome and X chromosome, except for chromosome 17. About one-third of the fat deposition QTLs were distributed on chromosome 7 and were mainly involved in the average backfat thickness, final rib backfat, tenth rib backfat, and subcutaneous shoulder fat thickness, which was basically consistent with the previous results. In addition, 8.3% (85/1028) QTLs were associated with intramuscular fat, and these QTLs were mainly located on chromosome 3 and distributed on chromosomes 2, 4, 6, 7, 9, 15, and X (Figure 5C). The above results show that the QTLs corresponding to DERs have abundant diversity at the chromosome level and the DERs are associated with intramuscular fat (Appendix A). 

### 3.7. Expression Regulation Analysis of DE lncRNAs, miRNAs, and circRNAs, and Their DE PTGs 

The results of gene co-expression showed that there are 23 DE lncRNAs and 18 DEGs in the lncRNA–mRNA network between the two groups. There were 9 DEGs and 11 DE miRNAs in the miRNA–mRNA network, and there was only one DEcircRNA and DEG between the two groups. The circRNA–lncRNA, circRNA–miRNA, and circRNA–mRNA networks are shown in Appendix A. 

Based on the ceRNA hypothesis, we analyzed the total transcriptome data and constructed the ceRNA regulatory network. The ceRNA network contained 4032 lncRNAs, 6785 mRNAs, and 815 circRNAs. By using a one-step neighbor network to construct different ceRNA combinations, we found that nine known miRNAs (ssc-miR-4334-3p, ssc-miR-339, ssc-miR-339-5p, ssc-miR-4331-3p, ssc-miR-671-5p, ssc-miR-874, ssc-miR-671-5p, ssc-miR-7138-3p, and ssc-miR-370) were involved in more relationship pairs in the mRNA–miRNA–lncRNA network, and may play a central role in the regulatory network. Similarly, we found nine miRNAs (ssc-miR-1343, ssc-miR-671-5p, ssc-miR-4331-3p, ssc-miR-328, ssc-miR-874, ssc-miR-9785-5p, ssc-miR-370, ssc-miR-1224, and ssc-miR-330) in the mRNA–miRNA–circRNA network that may play a central role in the regulatory network. There were 10 miRNAs in circRNA–miRNA–lncRNA network (ssc-miR-339, ssc-miR-339-5p, ssc-miR-4334-3p, ssc-miR-4331-3p, ssc-miR-370, ssc-miR-874, ssc-miR-574-5p, ssc-miR-1343,2320-5p, and ssc-miR-6782-3p). Thus, multiple miRNAs participate in a common regulatory role in the ternary regulatory network. CircRNA plays an important role in regulating gene expression by interacting with miRNA in mammals. We compared the relationships between DEmRNA and miRNA to obtain the DE circRNA–miRNA–DE mRNA interaction network (Figure 5A). 

We compared the relationships between DE mRNA and miRNA to obtain the DE circRNA–DE mRNA interaction network. Similarly, by using the DE lncRNA–miRNA relationship network, we obtained the DE lncRNA–miRNA–DE mRNA and the DE circRNA–miRNA–DE lncRNA interaction networks (Figure 6B,C). Nevertheless, the miRNAs in the ceRNA networks we constructed were not differentially expressed between the two groups.

Based on the integration analysis of key gene pathways in different ceRNA networks, Gene network analysis showed mitogen-activated protein kinase 10 (*MAPK10*), Janus kinase 1 (*JAK1*), signal transducer and activator of transcription 1 (*STAT1*), and other genes associated with fat deposition were enriched in the pathway; tyrosine kinase 2 (*TYK2*), interferon regulatory factor 9 (*IRF9*), fas associated via death domain (*FADD*), and other key genes are shown in Figure 6D. 

### 3.8. RNA Sequencing Results Validation Using qRT-PCR

To validate the accuracy of the RNA-Seq data, according to their expression levels, five RNAs were screened from the DEmRNAs and DE lncRNAs in low- and high-IMF groups. The genes Potassium Channel Regulator (*KCNRG*), HUS1 Checkpoint Clamp Component (*HUS1*), and lncRNA MSTRG.5761.2 were highly expressed in the high IMF group, while the genes *ACAA2* and lncRNA MSTRG.40179 were lowly expressed. We designed different primers and used cDNA as amplification template. QRT-PCR results showed that the expression levels of these candidate RNAs did not change significantly between low- and high-IMF group, which was consistent with our sequencing analysis, indicating that our estimation of abundance was accurate (Figure 7).

## 4. Discussion

IMF content is one of the polygenic traits in animals and is an important determinant of meat quality. Increasing the accumulation of intramuscular fat can promote the formation of meat marble patterns and improve the taste, flavor, color, and other characteristics of meat [2,3,29,30]. Therefore, in view of the importance of IMF to livestock production economics, it is of great significance to clarify the molecular mechanisms of IMF deposition [30,31]. Even in the same breed and under the same breeding conditions, genetic factors leading to individual accumulation of IMF content are different. Moreover, the association between genomic markers and IMF deposition is not always consistent, so it is essential to explore the potential molecular mechanisms related to IMF [32]. Up to now, some studies have identified candidate genes (protein-coding and noncoding genes) related to meat quality traits and used them in practical production [33,34,35]. Intramuscular fat is highly complex and metabolically active, which involves complex metabolic processes and pathways, and also involves multiple genes. However, the regulatory mechanism of fat deposition is poorly understood.

RNA-seq technology was used for transcriptome analysis of porcine LDM samples with different IMF contents. In total, 218 DEGs were identified between the two groups, many of which have known functions in lipid metabolism. For example, the adipogenic gene *SFRP4* can positively regulate the expression of adipogenic genes through the Wnt/β-catenin signaling pathway, thereby promoting the formation of fat [36,37]. *GPAT* is a rate-limiting enzyme involved in triglyceride synthesis [38]. *GPAT3* is the main form of *GPAT* expressed in adipocytes and plays a crucial role in fat formation [39]. *ACAA2* is a key enzyme in the fatty acid oxidation pathway, which regulates cell apoptosis and triglycerides, and plays an important role in fatty acid metabolism. At the same time, *ACAA2* can also promote the differentiation of preadipocytes into adipocytes through *PPAR*, thereby regulating intramuscular fat content [40]. *TGFB3* is a regulator of the number of adipocytes, which can increase the number of adipocytes in white adipose tissue (WAT) and reduce glucose tolerance [41]. *ACOX2* can also involve in the regulation of chicken IMF with different growth rates though PPAR pathway [42]. Significantly, we found that *SPP1* and *THBS4* were enriched in the PI3K-Akt signaling pathway involved in lipid metabolism [43,44]. *CXCL16* can participate in lipid metabolism by triggering downstream *PI3K, Akt*, and *IKK* signal transduction events. It can be seen that DEGs participate in multiple pathways at the same time, forming a complex regulatory network involved in fatty acid biosynthesis and metabolism. It is also noteworthy that among these known genes, *SFRP4*, *GPAT3*, and *ACAA2* were consistent with IMF content trends and play a positive regulatory role in intramuscular fat deposition. However, *ACOX2* showed the opposite trend, and the inconsistent expression trend may be related to some other potential gene regulation or gene tissue-specific expression. These are worthy of further study for understanding the complex regulatory mechanisms of intramuscular fat.

In this study, the number of lncRNAs identified in our results was significantly different from that in Duroc and Luchuan pigs (4868 lncRNAs) [45], Jinhua and Landrace pigs (4910 lncRNAs) [46], and Songliao and Landrace pigs (1071 lnRNAs) [47], which may be due to the rich genetic diversity of F2 resource pigs. The identified lncRNAs showed typical characteristics, such as a shorter transcript length, fewer exons, a longer exon length, and a lower expression level compared with protein-coding transcripts, which is consistent with previous studies [48]. In total, 274 PTGs were differentially expressed between low- and high-IMF pigs, and this indicated that some DE regulatory RNAs may contribute to the deposition and metabolism of IMF by regulating their PTGs. In addition, although muscle is an important metabolic tissue in pigs and is involved in a variety of muscle development events, such as muscle growth and lipid metabolism, we infer that some DERs in the LDM are related to muscle to a certain extent, but our research focus was on IMF-related RNA. The QTL analysis results of the DE lncRNAs showed that these were mostly located in the QTLs for IMF content, which further proved our speculation to some extent. This result is also consistent with a previous study [49]. Previous studies have shown that lncRNA can regulate gene expression in some ways, including cis- and trans-regulation [50,51,52]. In our study, we found that lncRNA and its adjacent genes showed a strong correlation. Functional annotation and network analyses showed that MSTRG.16111.1 and its target gene, *ACAA2*, were significantly downregulated. *ACAA2* can participate in PPAR signaling, and that the primary bile acid biosynthesis pathway was involved in lipid metabolism in muscle [53].

The DE miRNAs and DE circRNAs identified for IMF showed that they were significantly enriched in the lipid-related pathways, such as the glucagon signaling pathway, the mTOR signaling pathway and adherens junction. They were mainly involved in the insulin signaling pathway, the *MAPK* signaling pathway, the *PI3K-Akt* signaling pathway, and the *TGF-*β signaling pathway. Studies have shown that ssc-miR-208b may be essential for IMF metabolism [54], and ssc-miR-499-5p is associated with type I muscle fibers [55]. Ssc-miR-190b regulates lipid metabolism and insulin sensitivity by targeting *IGF-1* and *ADAMTS9* [56].

Although recent studies have reported that miRNAs are involved in the development of intramuscular preadipocytes [57,58], the molecular regulation mechanism of miRNAs in porcine IMF development remains largely unknown. Studies have shown that inhibition of ssc-miR-499-5p expression in nonalcoholic fatty liver disease (*NAFLD*) cells reduces lipid deposition and of triglyceride (TG) content [59]. However, in this study, ssc-miR-499-5p was upregulated in high-IMF animals, which may be due to the high conservation and tissue specificity of miRNA, which plays different roles in different tissues.

We constructed a ceRNA regulatory network using co-expressed and targeted RNAs. Although we constructed the DE circRNA–miRNA–DE mRNA, DE lncRNA–miRNA–DE mRNA, and DE circRNA–miRNA–DE lncRNA networks based on the “sponge adsorption” theory, the results showed that the genes were mainly involved in immune regulation and anti-infection. According to previous research results, these signaling pathways and key genes are also widely involved in lipid metabolism and fat deposition [60,61,62]. These may play an important role in the complex molecular process of intramuscular fat deposition. In this study, a key gene integration analysis of different combinations of ceRNA pairs was carried out. According to previous research results, these signaling pathways and key genes are also widely involved in lipid metabolism and fat deposition, but the enriched genes, including the *MAPK10* and *JAK/STAT* pathways, were related to lipid metabolism including fat deposition and fatty acid β oxidation [63]. At the same time, studies have shown that decreased *TYK2* and *STAT1* promoted the expression of *PPARγ* and *FAS* in adipose tissue [64,65]. *FADD* was recently reported as a key regulator of lipid metabolism, and *FADD* is a master regulator of glucose and fat metabolism [66].

## 5. Conclusions

In our study, we identified and analyzed mRNAs, miRNA, lincRNAs, and circRNAs between low- and high-IMF samples from the longissimus dorsi muscle (LDM) in a Large White × Min F2 resource population of pigs. In total, 290 RNAs and 527 PTGs were found to be differentially expressed between low- and high-IMF pigs. Function analysis indicated that many regulatory RNAs, such as MSTRG.1611.1, MSTRG.35593.1, MSTRG.77761.1, ssc-miR-208b, and ssc-miR-190b, may have contributed to the differences in the IMF-related processes. However, the function and molecular regulatory mechanisms between regulatory RNAs and their PTGs remain unclear and require further exploration.

## Figures and Tables

**Figure 1 animals-11-03212-f001:**
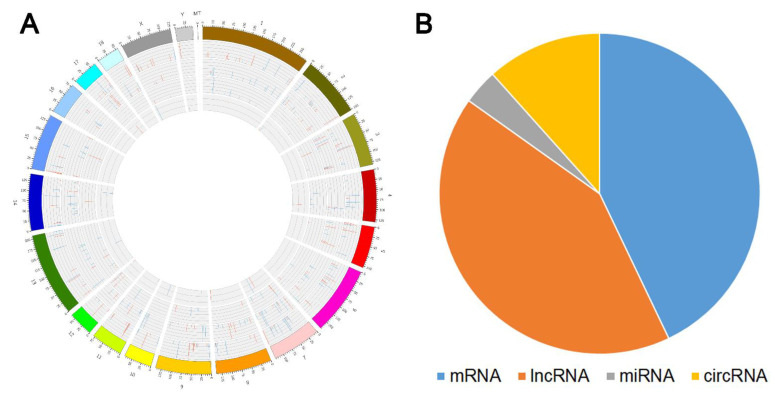
Statistics of differentially expressed RNAs in the high- and low- IMF groups. (**A**) The genome-wide distribution and expression schema for differentially expressed RNAs. (**B**) Number of differentially expressed RNAs in the two groups (*p*-value < 0.01 (mRNA, lncRNA, and circRNA), and *p*-value < 0.05 (miRNA)).

**Figure 2 animals-11-03212-f002:**
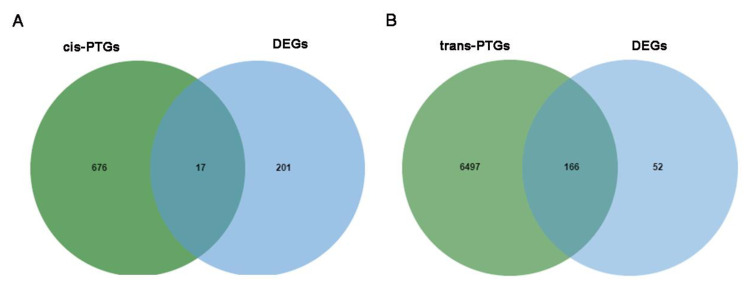
(**A**) The cis-regulated PTGs of lncRNAs that were differentially expressed between the two groups; 17 co-expressed genes from the DEGs and DE lncRNA target genes are shown. (**B**) The trans-regulated PTG differences between the two groups; 166 co-expressed genes from the DEGs and DE lncRNA target genes are shown.

**Figure 3 animals-11-03212-f003:**
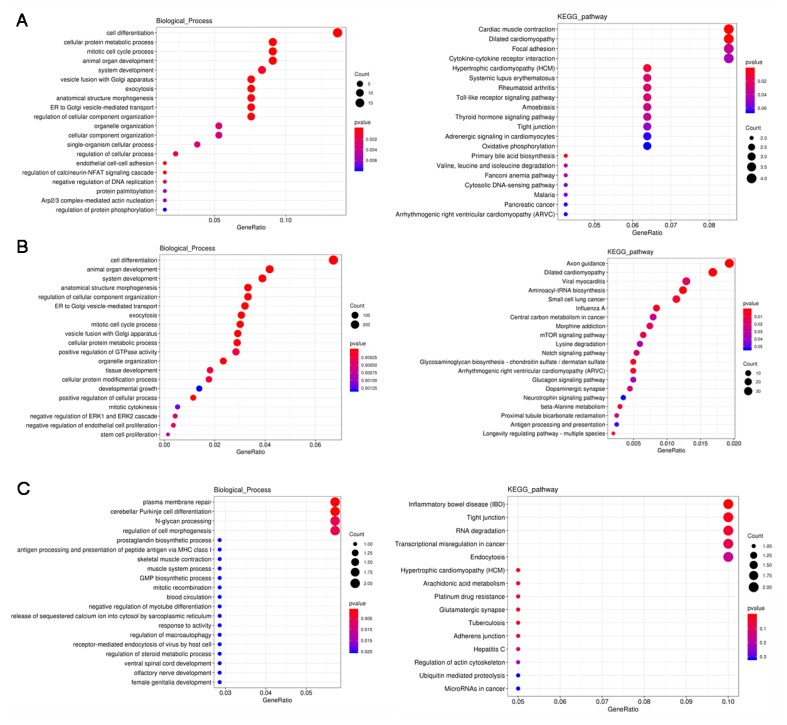
Gene ontology and pathway analysis of the DEGs and the potential target genes (PTGs) of DE miRNAs and circRNAs. (**A**) Gene ontology and pathway analysis of DEGs. (**B**) Gene ontology and pathway analysis of DE miRNA target genes. (**C**) Gene ontology and pathway analysis of the DE circRNA target genes.

**Figure 4 animals-11-03212-f004:**
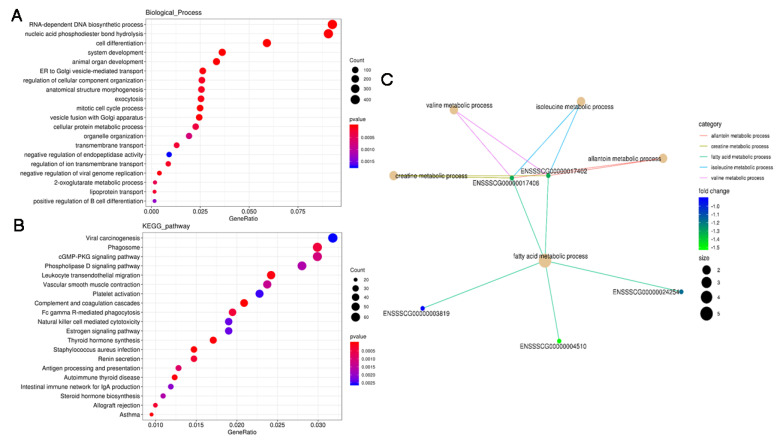
GO and KEGG pathway analysis of the PTGs of DE lncRNAs. (**A**) GO biological process analysis for all DE lncRNAs. (**B**) KEGG pathway analysis for all DE lncRNAs. (**C**) Gene network of PTGs enriched in fatty acid metabolism via *cis*-regulation.

**Figure 5 animals-11-03212-f005:**
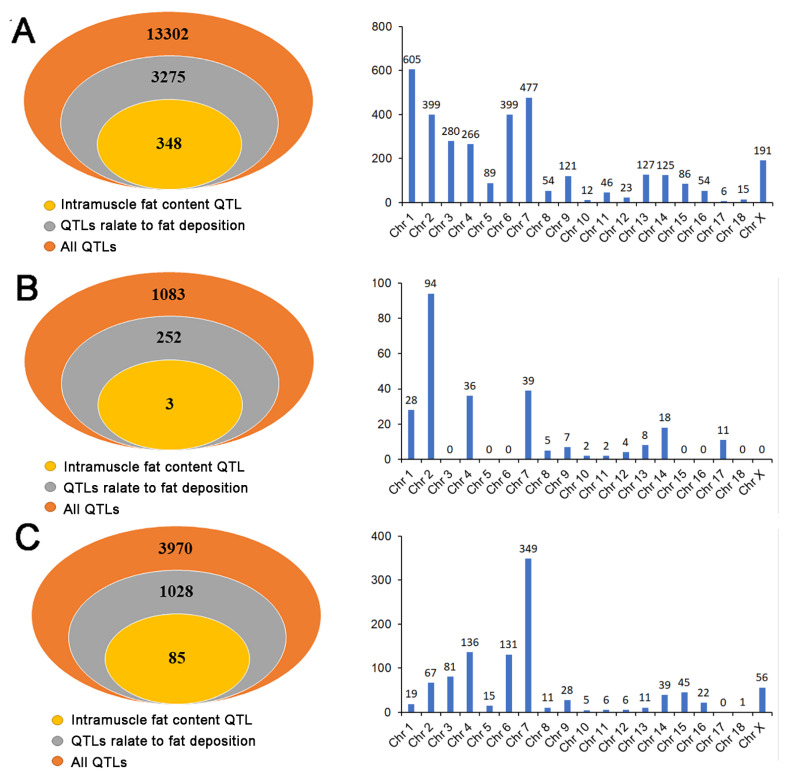
Quantitative trait locus analysis of DE RNAs. (**A**) The number distribution of QTLs related to fat deposition, the number of QTLs related to fat deposition, and the chromosome distribution of the QTLs related to fat deposition of the DE lncRNAs. (**B**) The number distribution of QTLs related to fat deposition, the number of QTLs related to fat deposition, and the chromosome distribution of the QTLs related to fat deposition of the DE miRNAs. (**C**) The number distribution of QTLs related to fat deposition, the number of QTLs related to fat deposition, and the chromosome distribution of the QTLs related to fat deposition of the DE circRNAs.

**Figure 6 animals-11-03212-f006:**
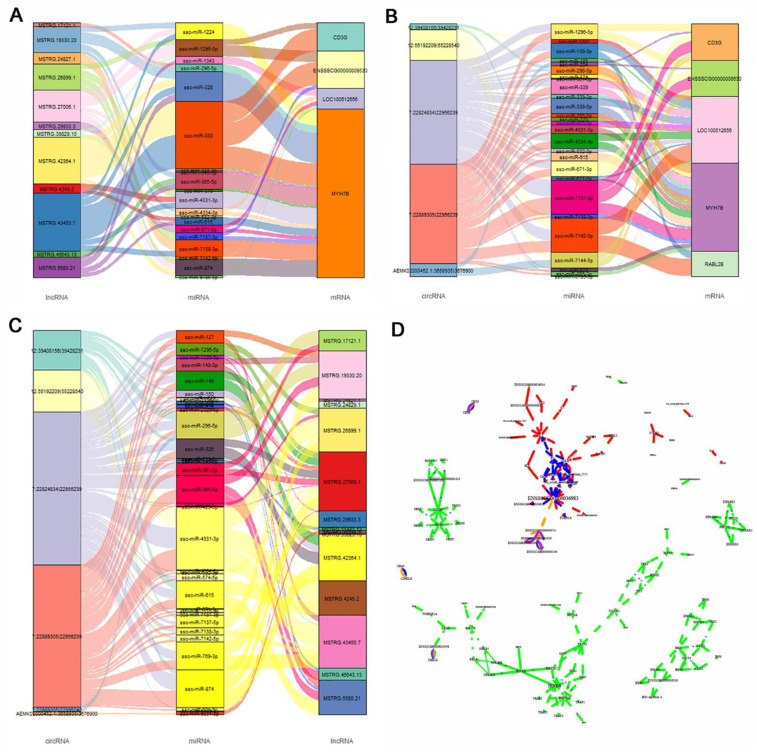
Co-expressed networks of differentially expressed (DE) mRNAs, lncRNAs, and circRNAs, and their targeted miRNAs. (**A**) Co-expressed networks of DE mRNAs and DE lncRNAs with the targeted miRNAs. (**B**) Co-expressed networks of DE mRNAs and DE circRNAs with the targeted miRNAs. (**C**) Co-expression networks of DE lncRNAs and DE circRNAs with the targeted miRNAs. (**D**). The networks of genes in the top five pathways, and the integration analysis of key gene pathways in different ceRNA networks.

**Figure 7 animals-11-03212-f007:**
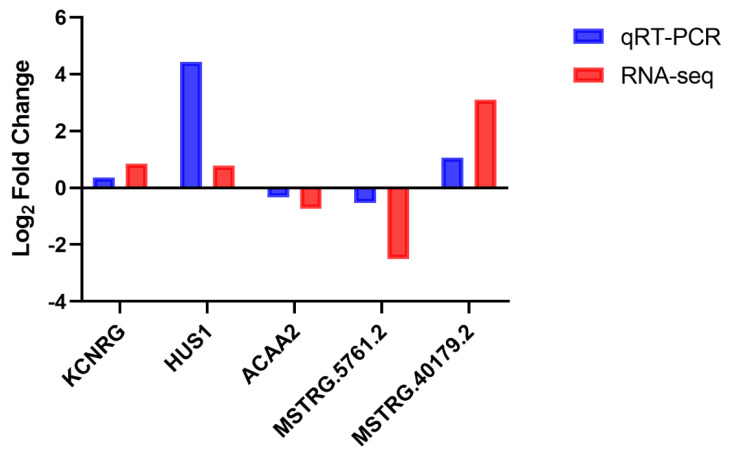
The RNA sequencing analysis data were verified by qRT-PCR. The relative expression levels of five differentially expressed RNAs were analyzed between RNA sequencing and qRT-PCR.

**Table 1 animals-11-03212-t001:** Description of IMF content between the two groups.

Sample	Carcass Weight (kg)	IMF (%)	Group
H1	82.6	4.07	High IMF
H2	113	4.40	High IMF
H3	113	4.56	High IMF
H4	97.4	4.98	High IMF
H5	96.8	5.43	High IMF
L1	92	1.05	Low IMF
L2	113	1.18	Low IMF
L3	125.6	1.28	Low IMF
L4	89.4	1.57	Low IMF
L5	94.4	1.60	Low IMF

**Table 2 animals-11-03212-t002:** Summary of top up-regulated and down regulated DE circRNAs, miRNAs, lncRNAs, and mRNAs between high- and low-IMF groups.

RNA	Regulated	log2FC	*p*-Value	Type
*MSTRG.19330.20*	up	6.573623	5.41 × 10^−^^62^	lncRNAs
*MSTRG.40179.2*	up	3.10731	1.32 × 10^−10^	
*MSTRG.44176.8*	up	1.908359	2.75 × 10^−5^	
*MSTRG.39829.10*	up	1.710734	4.15 × 10^−4^	
*MSTRG.38601.10*	up	1.476951	1.54 × 10^−6^	
*MSTRG.29140.1*	down	−1.94114	2.37 × 10^−5^	
*MSTRG.25219.1*	down	−1.99459	2.95 × 10^−5^	
*MSTRG.9199.1*	down	−2.01282	4.72 × 10^−6^	
*MSTRG.5761.2*	down	−2.50226	3.96 × 10^−8^	
*MSTRG.44725.16*	down	−2.71452	1.45 × 10^−10^	
*novel_miR_118*	up	1.459799	1.07 × 10^−2^	miRNAs
*ssc-miR-208b*	up	1.310524	7.14 × 10^−3^	
*novel_miR_398*	up	1.297012	3.44 × 10^−2^	
*novel_miR_278*	up	1.272563	3.34 × 10^−2^	
*ssc-miR-190b*	up	1.198435	1.12 × 10^−2^	
*ssc-miR-499-5p*	up	1.185263	1.01 × 10^−2^	
*novel_miR_185*	down	−1.54969	1.09 × 10^−2^	
*novel_miR_45*	down	−1.74161	3.76 × 10^−3^	
*novel_miR_476*	down	−1.78496	3.59 × 10^−3^	
*novel_miR_45*	down	−1.74161	3.76 × 10^−3^	
*novel_miR_476*	down	−1.78496	3.59 × 10^−3^	
*12:39408156|39428231*	up	9.005799	1.98 × 10^−6^	circRNAs
*14:71348983|71349948*	up	6.965969	4.78 × 10^−4^	
*3:44121881|44122061*	up	6.819521	3.29 × 10^−4^	
*9:125732918|125735258*	up	6.586107	1.03 × 10^−3^	
*13:71794794|71797638*	up	6.346968	1.25 × 10^−3^	
*1:108385212|108386218*	down	−5.96632	1.61 × 10^−3^	
*12:59320434|59323398*	down	−6.16306	1.81 × 10^−3^	
*7:68514625|68532510*	down	−6.37264	1.20 × 10^−3^	
*9:66405629|66409132*	down	−6.57113	8.32 × 10^−4^	
*4:50433434|50447885*	down	−6.83942	4.52 × 10^−4^	
*RDH16*	up	0.993121	1.20 × 10^−4^	mRNAs
*ENSSSCG00000045560*	up	0.928328	2.29 × 10^−4^	
*KCNRG*	up	0.856391	6.58 × 10^−4^	
*ENSSSCG00000045892*	up	0.844323	8.67 × 10^−4^	
*RABL2B*	up	0.831953	1.16 × 10^−3^	
*SPP1*	down	−0.92796	2.69 × 10^−4^	
*CIB2*	down	−0.93599	8.18 × 10^−5^	
*PTPMT1*	down	−0.96233	5.00 × 10^−5^	
*MYH7B*	down	−0.99074	1.05 × 10^−5^	
*GPNMB*	down	−1.15542	1.07 × 10^−5^	

*RDH16:* retinol dehydrogenase 16; *KCNRG*: potassium channel regulator; *RABL2B*: RAB, member of ras oncogene family similar to 2B; *SPP1*: Secreted Phosphoprotein 1; *CIB2*: calcium and integrin binding family member 2; *PTPMT1*: protein tyrosine phosphatase mitochondrial 1; *MYH7B*: myosin heavy chain 7B; and *GPNMB:* glycoprotein nmb.

## Data Availability

The data presented in this study are available in SRA with the project number of PRJNA776032.

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
