# Peer review of "Systematic Identification and Comparison of the Expressed Profiles of lncRNAs, miRNAs, circRNAs, and mRNAs with Associated Co-Expression Networks in Pigs with Low and High Intramuscular Fat"

_animals, 2021, doi:10.3390/ani11113212_

Round 1

Reviewer 1 Report

The manuscript is important for understanding the ncRNAs roles in the biology of lipid metabolism. The experiments were well designed. The selections of methods are suitable, and the methods are used appropriately. The introduction and discussion are concise.

I have some minor suggestions:

  1. It is not clear about the use of DEseq2 and EdgeR analyses; the authors should check it. Why was each software used, and how models were fitted?
  2. Why did the authors not using FDR or other methods for controlling of multiple testing for setting the threshold of P values?
  3. Some inconsistency in the names of miRNAs, abbreviations should be checked
  4. Some figures did not give the information why it might need to provide one table for a summary of DE results
  5. It is better to deposit the raw sequence data in the SRA or other public domain

Line 14: Write the full names for lncRNAs, miRNAs, and circRNAs as they are not standard abbreviations.

Line 16: How did the authors access the quality to claim that: high quality

Line 18: What do the authors mean by "idea"?

Line 24: Define abbreviation for these RNAs

Line 26: It is not clear PTGs of what (miRNAs or all types of ncRNAs).

Line 32-33: It is not surprising as the QTL for fat traits spans almost all pig chromosomes.

Line 38: I am not sure about the word development. Did the authors characterize the sample or tissue at different stages of development?

Line 64: Add the references for them; they are rare but existing

Line 115: Write a full name for ncRNA

Line 142-143: why did the authors use both DEseq and EdgeR? Which one used Deseq2 and which used EdgeR?

Line 143-144: Did the authors use the adjusted p values for threshold?

Line 148: K mean kilobase pairs?

Line 169: Pearson changes to Pearson’s

Line 170-171: What is about mRNA-miRNAs?

Line 202-203: The authors did not need to present all three levels of threshold; otherwise, explain why they needed three levels?
Line 213-214: Please deposit the sequence data in SRA or some public domain

Figure 1 C, D, E, F did not make any senses, remove them or add them in the supplementary

I suggest the author present the top DE gene and ncRNAs in the table.

Line 254: change differentially expressed to DE

Line 338: It is not association analysis; the authors simply find the overlapping regions

The quality of figure 6 is too low Figure 6 D could not be seen in the PDF version,

Line 418: The gene names should be in Italics; check it consistently throughout the manuscript.

Line 481: Please add more references after studies or change studies to study

Line 524-525: Why did the authors show only novel RNAs? I suggest removing them as they might not be true RNAs molecules.

In the discussion, please specify the species names for each miRNA (line 493, 500, etc.).

Author Response

1) It is not clear about the use of DEseq2 and EdgeR analyses; the authors should check it. Why was each software used, and how models were fitted?

Response: In order to mining enough differentially expressed RNAs (DERs), both DEseq2 and EdgeR software were used to do differential expression (DE) analysis. Negative binomial generalized linear models were fitted in both two software. As the DERs investigated by this two software are almost the same, we only select the results of DEseq2 to do further research. We have corrected the description of DEseq2 and EdgeR analyses to “DEseq2(v1.6.3) R package [21] with negative binomial generalized linear model was used to screen differentially expressed RNAs (DERs)” in current version of manuscript (Lines 153-154).

2) Why did the authors not using FDR or other methods for controlling of multiple testing for setting the threshold of P values?

Response: FDR is a powerful method for controlling multiple testing. But usually, the FDR criteria is a little strict. In this study, we first use FDR method to determine DERs, and found that the DERs are too little to do further analysis such as GO, KEGG and ceRNA network construction. So, we chose the P values to determine DERs. In some research such as Cardoso et al. (2017), Xie et al.(2020), and so on, statistical criteria of P-value ≤ 0.05 were used to determine DERs. And moreover, in our study, the validation study indicated that most of the DERs determined by P value could be validated by qRT-PCR.

Cardoso TF, Cánovas A, Canela-Xandri O, González-Prendes R, Amills M, Quintanilla R. RNA-seq based detection of differentially expressed genes in the skeletal muscle of Duroc pigs with distinct lipid profiles. Sci Rep. 2017 Feb 14;7:40005. doi: 10.1038/srep40005.

Xie Y, Li J, Li P, Li N, Zhang Y, Binang H, Zhao Y, Duan W, Chen Y, Wang Y, Du L, Wang C. RNA-Seq Profiling of Serum Exosomal Circular RNAs Reveals Circ-PNN as a Potential Biomarker for Human Colorectal Cancer. Front Oncol. 2020 Jun 18;10:982. doi: 10.3389/fonc.2020.00982.

3) Some inconsistency in the names of miRNAs, abbreviations should be checked

Response: We have checked the names of miRNAs in all occurrences throughout the manuscript and unified all the name to miRNAs (Figure1 and Line 75,110,156).

4) Some figures did not give the information why it might need to provide one table for a summary of DE results

Response: Follow your suggestion we have checked all the figures and added one table for a summary of DE results under Figure 1. (Line 251).

5) It is better to deposit the raw sequence data in the SRA or other public domain

Response: We have submitted the raw data to SRA with the project number of PRJNA776032 (Lines 590-591).

6) Line 14: Write the full names for lncRNAs, miRNAs, and circRNAs as they are not standard abbreviations.

Response: Done (Lines 14-15)

7) Line 16: How did the authors access the quality to claim that: high quality

Response: We have changed “high quality” to “high throughput”. (Line 17)

8) Line 18: What do the authors mean by "idea"?

Response: We have deleted this word. (Line 19)

9) Line 24: Define abbreviation for these RNAs

Response: Done (Lines 26-27)

10) Line 26: It is not clear PTGs of what (miRNAs or all types of ncRNAs).

Response: Done (Lines 31-32)

11) Line 32-33: It is not surprising as the QTL for fat traits spans almost all pig chromosomes.

Response: We have deleted the description of overlapped QTLs for fat traits and only kept the description of overlapped QTLs for IMF (Line 37-38)

12) Line 38: I am not sure about the word development. Did the authors characterize the sample or tissue at different stages of development?

Response: The word development should be “formation”, and we have revised in the manuscript (Line 43)

13) Line 64: Add the references for them; they are rare but existing

Response: Done (Line 71)

14) Line 115: Write a full name for ncRNA

Response: Done (Lines 125-126)

15) Line 142-143: why did the authors use both DEseq and EdgeR? Which one used Deseq2 and which used EdgeR?

Response: In order to mining enough differentially expressed RNAs (DERs), both DEseq2 and EdgeR software were used to do differential expression (DE) analysis. Negative binomial generalized linear models were used in both two software. As the DERs investigated by this two software are almost the same, we only select the results of DEseq2 to do further research. We have corrected the description of DEseq2 and EdgeR analyses to “DEseq2(v1.6.3) R package [21] with negative binomial generalized linear model was used to screen differentially expressed RNAs (DERs)” in current version of manuscript (Lines 153-154).

16) Line 143-144: Did the authors use the adjusted p values for threshold?

Response: P adjust is a powerful but strict. In this study, we first use P adjust to determine DERs, and found that the DERs are too little to do further analysis such as GO, KEGG and ceRNA network construction. So, we chose the P values to determine DERs. In some research such as Cardoso et al. (2017), Xie et al.(2020), and so on, statistical criteria of P-value ≤ 0.05 were used to determine DERs. And moreover, in our study, the validation study indicated that most of the DERs determined by P value could be validated by qRT-PCR. So, we use the P value for threshold in this study. (Lines 154-157)

17) Line 148: K mean kilobase pairs?

Response: Yes, we have changed this word. (Line 161)

18) Line 169: Pearson changes to Pearson’s

Response: Done (Line 183)

19) Line 170-171: What is about mRNA-miRNAs?

Response: mRNA-miRNAs referred to mRNA-miRNA pairs, which we have revised. (Line 185)

Response: Done (Line 196)

21) Line 202-203: The authors did not need to present all three levels of threshold; otherwise, explain why they needed three levels?

Response: In our analysis, P < 0.05 was considered to indicate a significant difference, P < 0.01 and P < 0.001 indicated extremely significant differences. We have added this explanation in the manuscript. (Lines 216-218)

22) Line 213-214: Please deposit the sequence data in SRA or some public domain

Response: We have submitted the raw data to SRA with the project number of PRJNA776032 (Lines 590-591).

23) Figure 1 C, D, E, F did not make any senses, remove them or add them in the supplementary

Response: We have removed Figure 1 C, D, E, F. (Line 245)

24) I suggest the author present the top DE gene and ncRNAs in the table.

Response: Follow your suggestion, we have presented the top DE gene and ncRNAs in Table 2. (Line 254)

25) Line 254: change differentially expressed to DE

Response: Done (Line 288)

26) Line 338: It is not association analysis; the authors simply find the overlapping regions

Response: We have changed the description to overlapping analysis. (Line 372)

27) The quality of figure 6 is too low Figure 6 D could not be seen in the PDF version,

Response: We have replaced Figure 6D with a high-quality version. (Figure 6D)

28) Line 418: The gene names should be in Italics; check it consistently throughout the manuscript.

Response: We have checked the gene names throughout the manuscript and made necessary modification. (LineS 305-313,359, 450,456,494-498,528)

29) Line 481: Please add more references after studies or change studies to study

Response: Follow your suggestion, we have changed studies to study. (Line 524)

30) Line 524-525: Why did the authors show only novel RNAs? I suggest removing them as they might not be true RNAs molecules.

Response:  Follow your suggestion, we have removed the novel RNAs and added two known RNAs. (Line 568-569).

31) In the discussion, please specify the species names for each miRNA (line 493, 500, etc.).

Response: Done (Lines 535-537).

Reviewer 2 Report

The article entitled "Systematic Identification and Comparison of the Expressed Profiles of lncRNAs, miRNAs, circRNAs and mRNAs with Associated Co-expression Networks in Pigs with Low and High Intramuscular Fat" represents very actual trend in genomic investigations. Authors correctly chose breeds with different IMF content and then 10 animals with its extremely values. Applied methods are high-throughput and also well chosen, similar to bioinformatics analysis.

I have found, however some errors and inaccuracies which should be corrected before publishing.

Some of them are mentioned below:

  • Each shortcut should be explained when is first used and not repeat e.g. lines 12, 20, 44 for IMF lines 31, 53 and many more for QTL, lack of explanation of shortcuts in line 14 - lncRNA and others, line 34 - ceRNA, DE is mentioned in line 33, shortcut explained in line 71, similar PTGs in lines 28 and 70.
  • Replace "and so on" by "and many more" in line 13.
  • Authors should be consistent with the order of RNA types e.g. lines 14 and 24 and keep it in the whole text also when present results, give Set the one order at the beginning of text e.g. mRNA, miRNA, lncRNA, circRNA and follow it.
  • Line 46 - should be "meat quality" not "muscle quality"
  • Line 57 - shortcut for GWAS is Genome-wide association study
  • Line 79 - All animal experiments in our research were carried out with the ethical...
  • Line 86 - Pigs were weighed and slaughtered in a commercial slaughterhouse.
  • Line 115 an 117 - explain shotcuts for ncRNAs and sRNA.
  • Line 151 - separate sentences: "... with the expression of lncRNAs. The absolute Pearson’s coefficient (r)..."
  • Lines 207-209 - move the bracket in the correct place: " The individuals in the study were from the F2 population, with an average IMF content of 2.85 ± 1.83 (high-IMF group: 4.07 < IMF < 5.43; low-IMF group: 1.05 < IMF < 1.60) and an average carcass weight of 109.31 ± 16.07.
  • Lines 209 - 210 - specify significant differences (what they relate to)
  • Figure 1 B - what is sRNA? Is it DE miRNA at p<0.05 (line 237)? If yes maybe p should be given in figure and include 99 DE circRNA at p<0.05.
  • Use small letters for RNA types e.g Fig. 1 D, lines 152, 239, 369.
  • Line 241 - put the shortcut: "...dependent cis- and trans-algorithms were used to predict the target genes (PTGs)..." which allow to use it in next sentences.
  • Line 261 - " GO analysis showed that the DE RNAs were mainly involved in the cell part of the cell component classification" Is it mean all types of RNA? State " cell part of the cell component classification" is not clear
  • Lines 263-264 - "Molecular function evaluation showed that the DERs were associated with binding" this sentence is too general.
  • Section 3.4. should be corrected, because sometimes text does not reflect what is in Figures. Follow the established RNA type order! Describe results as in lines 294-300! Decide to use GO numbers or not in whole section!
  • Rewrite the sentence: " In the cytokine–cytokine receptor interaction pathway, chemotherapeutic factors (such as CCL4), transforming growth factor (such as TGFB3) and chemokines (such as CXCL10, CXCL16) were highly expressed in low-IMF individuals." I propose: " Genes in the cytokine–cytokine receptor interaction, chemotherapeutic factors (such as CCL4), transforming growth factor (such as TGFB3) and chemokines (such as CXCL10, CXCL16) pathways were highly expressed in low-IMF individuals."
  • Use the italics for gene symbols e.g. line 281 - ACOX2
  • Figure 5. - graphs A and C are ordered by chromosome number, however graph B by QTL number per chromosome - standardize it .
  • Lack of C explanation under Figure 5.
  • Line 389 - cyclic RNA?
  • Lines 411-413 - standardize gene symbols with capitols and italics.
  • Figure 6. ABC - some names of RNA types are not visible even by high magnitude - change resolution or figure arrangement if possible. D is completely invisible - i think gene symbols should be given. For me it is not informative figure, so delete it or replace by better quality.
  • Line 436 - change the bracket for "(protein-coding and noncoding genes)"
  • Lines 437-438 - "Intramuscular fat is a highly complex and metabolically active tissue"
  • Is intramuscular fat a tissue?
  • Line 454 - "The fatty acid β-oxidation gene ACOX2 inhibits ACOX2 in human.." Gene inhibits itself? Do You mean via miRNA? Explain it, please.
  • Line 477 - "muscle development environments" is not correct term in this sentence.

Author Response

1) Each shortcut should be explained when is first used and not repeat e.g. lines 12, 20, 44 for IMF lines 31, 53 and many more for QTL, lack of explanation of shortcuts in line 14 - lncRNA and others, line 34 - ceRNA, DE is mentioned in line 33, shortcut explained in line 71, similar PTGs in lines 28 and 70.

Response: Thanks for your suggestion. Simple summary, abstract and main text are different parts, so the shortcut in these parts were repeated. Reviewer 1 also suggested us explain the whole name of shortcuts in different part. So, we did not change this in current version of manuscript.

2) Replace "and so on" by "and many more" in line 13.

Response: Done (Line 13)

3) Authors should be consistent with the order of RNA types e.g. lines 14 and 24 and keep it in the whole text also when present results, give Set the one order at the beginning of text e.g. mRNA, miRNA, lncRNA, circRNA and follow it.

Response: We have checked this throughout the manuscript and made corresponding modification. (Lines 14-15, 26-27, 31, 67-68, 75-76, 156)

4) Line 46 - should be "meat quality" not "muscle quality"

Response: Yes, we have revised it. (Line 52)

5) Line 57 - shortcut for GWAS is Genome-wide association study

Response: Follow your suggestion, we have changed the shortcut for GWAS. (Line 63)

6) Line 79 - All animal experiments in our research were carried out with the ethical...

Response: Done (Line 88)

7) Line 86 - Pigs were weighed and slaughtered in a commercial slaughterhouse.

Response: Done (Line 95)

8) Line 115 an 117 - explain shotcuts for ncRNAs and sRNA.

Response: Done (Lines 125, 127)

9) Line 151 - separate sentences: "... with the expression of lncRNAs. The absolute Pearson’s coefficient (r)..."

Response: Done (Line 164)

10) Lines 207-209 - move the bracket in the correct place: " The individuals in the study were from the F2 population, with an average IMF content of 2.85 ± 1.83 (high-IMF group: 4.07 < IMF < 5.43; low-IMF group: 1.05 < IMF < 1.60) and an average carcass weight of 109.31 ± 16.07.

Response: Done (Lines 222-226)

11) Lines 209 - 210 - specify significant differences (what they relate to)

Response: It is related to IMF. Done (Line 226)

12) Figure 1 B - what is sRNA? Is it DE miRNA at p<0.05 (line 237)? If yes maybe p should be given in figure and include 99 DE circRNA at p<0.05.

Response: In Figure 1B, sRNA is DE miRNA at p<0.05, and the number of DE circRNAs is 59. Corresponding modification have been done in Figure 1B. (Line 247, 251-252)

13) Use small letters for RNA types e.g Fig. 1 D, lines 152, 239, 369.

Response: Follow the suggestion of reviewer 1. Fig.1 D has been deleted. Other request modification has been done. (Line 267)

14) Line 241 - put the shortcut: "...dependent cis- and trans-algorithms were used to predict the target genes (PTGs)..." which allow to use it in next sentences.

Response: Done (Line 267)

15) Line 261 - " GO analysis showed that the DE RNAs were mainly involved in the cell part of the cell component classification" Is it mean all types of RNA? State " cell part of the cell component classification" is not clear

Response: In our study, all type of RNA were mainly involved in the cell part of cellular component. The classification of GO includes biological processes, cellular components, and molecular functions. Corresponding modification have done. (Line 290)

16) Lines 263-264 - "Molecular function evaluation showed that the DERs were associated with binding" this sentence is too general.

Response: We have deleted this sentence in current version of manuscript.

17) Section 3.4. should be corrected, because sometimes text does not reflect what is in Figures. Follow the established RNA type order! Describe results as in lines 294-300! Decide to use GO numbers or not in whole section!

Response: Follow your suggestion, section 3.4 has been divided to two section (section 3.5 and 3.6). We also have deleted the GO numbers, and have modified the results as the description in lines 294-300 (now in lines 326-332.). (Lines 289-332).

18) Rewrite the sentence: " In the cytokine–cytokine receptor interaction pathway, chemotherapeutic factors (such as CCL4), transforming growth factor (such as TGFB3) and chemokines (such as CXCL10, CXCL16) were highly expressed in low-IMF individuals." I propose: " Genes in the cytokine–cytokine receptor interaction, chemotherapeutic factors (such as CCL4), transforming growth factor (such as TGFB3) and chemokines (such as CXCL10, CXCL16) pathways were highly expressed in low-IMF individuals."

Response: Done (Lines 304-309)

19) Use the italics for gene symbols e.g. line 281 - ACOX2

Response: Done (Line 312)

20) Figure 5. - graphs A and C are ordered by chromosome number, however graph B by QTL number per chromosome - standardize it .

Response: Done (Line 383)

21) Lack of C explanation under Figure 5.

Response: Done (Line 388-390)

22) Line 389 - cyclic RNA?

Response: It should be circRNA. (Line 427)

23) Lines 411-413 - standardize gene symbols with capitols and italics.

Response: Done (Line 450-451)

24) Figure 6. ABC - some names of RNA types are not visible even by high magnitude - change resolution or figure arrangement if possible. D is completely invisible - i think gene symbols should be given. For me it is not informative figure, so delete it or replace by better quality.

Response: The resolution of Figure 6 ABC could not be changed. We have replaced Figure 6D with a high-quality version. (Line 432)

25) Line 436 - change the bracket for "(protein-coding and noncoding genes)"

Response: Done (Line 475)

26) Lines 437-438 - "Intramuscular fat is a highly complex and metabolically active tissue"

Is intramuscular fat a tissue?

Response: We have changed this sentence to “Intramuscular fat is highly complex and metabolically active” Done (Line 476-477)

27) Line 454 - "The fatty acid β-oxidation gene ACOX2 inhibits ACOX2 in human.." Gene inhibits itself? Do You mean via miRNA? Explain it, please.

Response: We have replaced this reference by a more suitable reference. (Line 495-496)

28) Line 477 - "muscle development environments" is not correct term in this sentence.

Response: We have replaced environments to events. (Line 519)

Round 2

Reviewer 1 Report

My comments have been addressed. 

Author Response

My comments have been addressed

Response: Thanks for your careful review of our paper. We have employed the  English language editing service by MDPI (NO. 35451), and have invited a native speaker to help us improving the English of our manuscript.

Reviewer 2 Report

Text of the manuscript has been significantly improved, however i have some recommendations:

lines 14, 25, 70 - should be messenger RNA

line 12 - remove additional space between is and key

line 23 - add were "created" or "constructed"

line 24 - i think Large White x Min can be deleted bacause was mentioned in the same sentence

move shorcut explanation (mRNA) from line 70 to 62

line 157 - lncRNAs (not LncRNA) and miRNA should be italicized

Author Response

Reviewer 2:

Text of the manuscript has been significantly improved, however I have some recommendations:

1)lines 14, 25, 70 - should be messenger RNA

Response: Done. (Lines 14,26, 67)

2)line 12 - remove additional space between is and key

Response: Done. (Line 12)

3)line 23 - add were "created" or "constructed"

Response: Follow your suggestion, we have added “constructed” in the manuscript (Line 24)

4)line 24 - i think Large White x Min can be deleted bacause was mentioned in the same sentence

Response: Follow your suggestion, we have deleted “Large White x Min”. (Line 26)

5)move shorcut explanation (mRNA) from line 70 to 62

Response: Done. (Line 67)

6)line 157 - lncRNAs (not LncRNA) and miRNA should be italicized

Response: Done. (Line 164)